# Slicing Vision Transformer for Flexible Inference

Yitian Zhang[1,2*]  Huseyin Coskun[1]  Xu Ma[2]  Huan Wang[2]  Ke Ma[1]
Xi (Stephen) Chen[1]  Derek Hao Hu[3]  Yun Fu[2]
[1]Snap Inc.   [2]Northeastern University   [3]Meta

## Abstract

Vision Transformers (ViT) is known for its scalability. In this work, we target to scale down a ViT to fit in an environment with dynamic-changing resource constraints. We observe that smaller ViTs are intrinsically the sub-networks of a larger ViT with different widths. Thus, we propose a general framework, named **Scala**, to enable a single network to represent multiple smaller ViTs with flexible inference capability, which aligns with the inherent design of ViT to vary from widths. Concretely, Scala activates several subnets during training, introduces Isolated Activation to disentangle the smallest sub-network from other subnets, and leverages Scale Coordination to ensure each sub-network receives simplified, steady, and accurate learning objectives. Comprehensive empirical validations on different tasks demonstrate that with only one-shot training, Scala learns slimmable representation without modifying the original ViT structure and matches the performance of Separate Training. Compared with the prior art, Scala achieves an average improvement of 1.6% on ImageNet-1K with fewer parameters. Code is available at here.

## 1 Introduction

Vision Transformers (ViTs) [9] are renowned for its scalability and various avenues [41, 5, 7] have been explored to scale up ViT models. To tailor ViTs to run on devices with limited resources, some recent progress [36, 35] utilize knowledge distillation [13] to scale down ViT. Particularly, DeiT [29] introduces two smaller variants of DeiT-B: DeiT-Ti and DeiT-S which have been widely used in resource-limited applications. Although these small ViTs exhibit enhanced efficiency, they lack the flexibility to implement customized adjustments that accommodate dynamically changing resource constraints in real-world scenarios, e.g., the computation budget of mobile phones depends on the energy level (low-power mode) and number of running apps. Consequently, the standard Separate Training (ST) protocol trains models with different sizes separately to provide a spectrum of options with diversified performance and computation. ST requires repetitive training procedures to produce multiple model choices, and the challenge is amplified for foundation models [26, 24, 17]. From users' perspective, they are only offered limited model choices, that might not cater to all scenarios.

Analyzing the architectures of ViT-Ti/S/B, we observe that these ViTs are the same architecture with the only difference in the number of embedding dimensions (we ignore the difference in the number of heads as it does not impact the overall model size), indicating smaller ViTs are intrinsically the sub-networks of larger model with different widths (see Fig. 1). This suggests that a large ViT can be transformed to represent small models by uniformly slicing the weight matrix at each layer. Given a width ratio $r$, we adjust the size of the network by this single hyperparameter, allowing a single ViT to represent multiple small variants with the weights of those sub-networks shared in a nested nature, e.g., ViT-B ($r$=0.25) equals ViT-Ti and ViT-B ($r$=0.5) corresponds to ViT-S. In this manner, we empower ViTs for flexible inference capability, and we aim to slice a ViT within a broad slicing

---

*Work done when Yitian was an intern at Snap Inc.

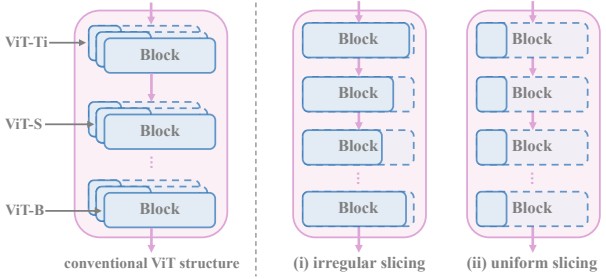

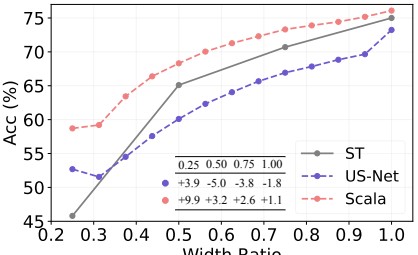

Figure 1: Illustration of different means to slice the ViT architecture. Irregular slicing [2, 38, 4] results in unconventional structures while uniform slicing [37] aligns with the inherent design of ViT to vary from widths.

Figure 2: The available uniform slicing method US-Net [37] lags behind Separate Training (ST) remarkably on ViTs. Performance gaps with ST are shown.

bound and fine-grained slicing granularity so that the diversity and number of sub-networks can be ensured for higher flexibility. This problem is non-trivial as fully training all the sub-networks within a constrained budget is nearly infeasible. Consequently, it is quite challenging for these subnets to match the performance of separate training.

Although various approaches have delved into slicing deep networks for flexible inference, the problem we target to resolve, i.e., uniformly slicing ViTs within a large slicing bound and fine-grained slicing granularity, is intrinsically different from others in three perspectives: (1) slicing strategy: as shown in Fig. 1 (i), the supernet training techniques [2, 38, 4] in NAS usually slice through multiple dimensions with a small slicing bound, resulting in irregularities in model architectures and a minor computational adjustment space. (2) slicing granularity: recent width slicing approaches [14, 18] either slice specific portions of the network or utilize a considerably large slicing granularity, leading to a limited number of models produced. (3) network architecture: US-Net [37] shares a similar vision with us but it has only demonstrated success in the CNN architecture.

It is crucial to note the fundamental differences between slicing CNN and ViT: (1) vanilla small ViTs such as ViT-Ti/S/B, are inherently designed to vary based on widths, aligning with our approach. Conversely, many CNNs are structured to vary from depths, like ResNet-18/34 [12], and slicing them by width brings unconventional architectures. (2) slimmable CNN necessitates calibration [39, 37] for each sub-network pre-inference due to Batch Normalization [15], unlike slimmable ViTs that can be directly utilized for evaluation. (3) the transformer architecture [33] has wider applications than CNN in this era, e.g., MAE [11], CLIP [26], DINOv2 [24], LLMs [31, 32, 1]. Nevertheless, ViTs have much less image-specific inductive bias than CNN and their slimmable ability remains unclear. As shown in Fig. 2, we empirically implement US-Net on ViT-S and observe substantial performance gaps at most width ratios compared to ST, indicating that the available solution of uniform slicing does not work well on the transformer architecture.

To investigate the underlying causes of this phenomenon, we conduct analyses in Sec. 3 which are briefly summarized in two folds: (1) ViTs display minimal interpolation ability, indicating that the optimization of intermediate subnets falls notably short compared to Separate Training (ST); (2) sustained activation of the smallest sub-network poses a negative effect on other subnets, which affects the overall performance as their weights are shared in a nested nature. To resolve these issues, we propose a general framework, named *Scala*, to enforce ViTs to learn slimmable representation. Specifically, we propose Isolated Activation to disentangle the representation of the smallest sub-network from other subnets while still preserving the lower bound performance. Besides, we present Scale Coordination to ensure each subnet receives simplified, steady, and accurate learning objectives. In this manner, the slimmable ViT can be transformed into multiple smaller variants during inference and match the performance of ST.

Compared to ST which trains all the subnets individually, Scala reduces the storage and training costs remarkably since the weights of smaller ViTs are shared with the full model and we only need one-shot training without extending the training duration. Further, Scala has a very large slicing bound and fine-grained slicing granularity, enabling diverse sub-network choices during evaluation. In this way, the delivered system can make tailored adjustments that accommodate dynamically changing resource constraints in real-world scenarios, promising the application on edge devices. Compared with the prior art SN-Net [25] which supports flexible inference on ViTs, Scala clearly outperforms it under different computation with fewer parameters. Moreover, Scala matches the

performance of ST on various tasks without modifying the network architecture, demonstrating its generalizability and potential to replace ST as a new training paradigm. The contributions are summarized as follows:

- Although slicing ViTs exhibits multiple advantages, we provide detailed analysis and practical insights into the slimmable ability between different architectures (Sec. 3 and Tab. 2) and find slicing the ViT architecture to be the most challenging problem.
- We propose a general framework *Scala* to enable ViTs to learn slimmable representation for flexible inference. We present Isolated Activation to disentangle the representation of the smallest subnet and Scale Coordination to ensure each subnet receives simplified, steady, and accurate signals.
- Comprehensive experiments on different tasks demonstrate that Scala, requiring only one-shot training, outperforms prior art and matches the performance of ST, substantially reduces the memory requirements of storing multiple models.

## 2   Related Work

**Scaling Up ViTs.** Like Transfromer [33] in NLP, scalability and performance improvements in ViTs [9] have been a central focus of recent research. Specifically, strategies have been explored to scale the depth of ViT [46, 30] and it is scaled to even larger sizes with almost 2 billion parameters and reaches new state-of-the-art results [41]. Afterward, ViTs have been scaled up to 4 billion [5] and 22 billion [7] parameters with extraordinary performance and enormous costs.

**Scaling Down ViTs.** The advent of ViTs has also sparked interest in scaling down these models. Techniques such as knowledge distillation [13] have been explored to reduce the ViT model size [36, 35]. For example, DeiT [29] presents smaller ViTs with 5M parameters. Additionally, researchers have explored quantization methods [22, 21] to further compress ViTs for deployment on edge devices. Unfortunately, these static models cannot make customized adjustments for resource-changing environments in real scenarios.

**Slimmable Neural Network.** The derivation of multiple smaller models from a single network has been previously explored but most works focus on the CNN structure. Slimmable Networks [39, 37] and its variants [34, 3, 10] train a shared network which adapts the width to accommodate the resource constraints during inference. Later, this idea is adapted into two-stage NAS methods [2, 38, 4] for supernet training. The supernet is scaled at multiple dimensions with a small computation change, in contrast to our work where we only scale the width dimension with a large slicing bound. SN-Net [25] is a recently proposed method that constructs a supernet with several pre-trained models and inserts linear layers to build dynamic routes for flexible inference. Recently, several of these techniques have been extended to Transformer architecture [14, 18], while they either scale part of the network or the slicing granularity is large which means they could only deliver very few models in the end. Differing from the previous works, our method is the first work to scale the ViT structure with large slicing bound and small slicing granularity which is intrinsically a more challenging problem.

## 3   Revisiting Slicing in Vision Transformer

Due to the excessive costs of constantly activating all the sub-networks during training, the sandwich rule is proposed in US-Net [37] to train the slimmable network at the smallest width, largest width, and 2 random intermediate widths in each iteration so that the performance of the lower bound and upper bound are guaranteed. Although the intermediate sub-networks are optimized less frequently compared to Separate Training (ST), US-Net manages to achieve comparable performance with ST on the CNN architecture. To have a better understanding of the distinction between CNN and ViT, we apply US-Net to MobileNetV2 [28], a CNN, and DeiT-S [29], a ViT, but constantly activate four sub-networks with the width ratio of $\{0.25, 0.5, 0.75, 1.0\}$ at each iteration. Subsequently, we evaluate the pre-trained models at both inbound $[0.25, 1.0]$ and outbound $(0, 0.25)$ unseen width ratios to evaluate their interpolation and extrapolation abilities, respectively. Shown in Fig. 3, CNN exhibits moderate interpolation and extrapolation capabilities by achieving acceptable performance at previously unobserved widths during training. In stark contrast, ViT fails entirely at unseen widths, suggesting that optimizing larger sub-networks does not directly benefit the performance of smaller ViTs, even though their weights are shared in a nested nature.

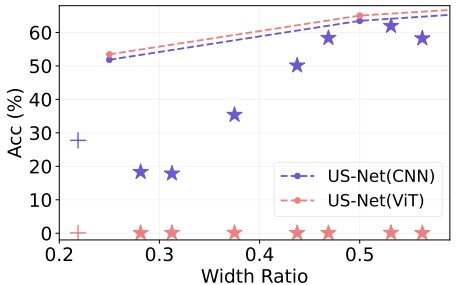
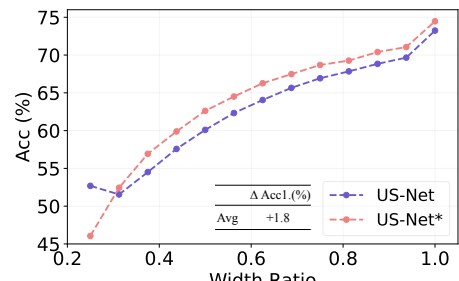

Figure 3: Evaluating US-Net over CNN and ViT at unseen width ratios to examine the interpolation (denoted as ⋆) and extrapolation (denoted as +) abilities on ImageNet-1K.

Figure 4: We train US-Net on ViT-S without constantly activating the smallest subnet (denoted as *) and observe an average performance gain of 1.8% at other width ratios on ImageNet-1K.

We analyze the results from the expected training epochs for each sub-network. Let $X$ represent the total number of networks to be delivered, wherein $X - 2$ intermediate sub-networks are included, and according to the sandwich rule, two of these are randomly sampled during each iteration. Formally, the expected training epochs for the intermediate networks $\xi$ can be expressed as:

$$\xi = \frac{2}{X - 2} \times \eta, \tag{1}$$

where $\eta$ is the number of training epochs for the full model and it suggests that the optimization of most sub-networks falls notably short compared to ST. As ViT has demonstrated minimal interpolation ability at unseen widths compared to CNN, each sub-network within the slimmable ViT requires optimal utilization of every training iteration to achieve satisfactory performance. Nevertheless, the smallest subnet is constantly activated during training according to the sandwich rule and we hypothesize that the over-emphasis of the smallest sub-network, often exhibits the worse performance, may increase the training difficulty of other subnets as their weights are shared in a nested nature. To validate it, we implement US-Net [37] on DeiT-S without constantly activating the smallest sub-network. Fig. 4 verifies our hypothesis showing an accuracy drop at the smallest subnet but a significant performance improvement at other width ratios.

## 4   Scala

We first introduce the training and inference paradigms of Scala. Then, we describe Isolated Activation which disentangles the smallest subnet from other sub-networks while maintaining the lower bound performance. Further, we present Scale Coordination to ensure each subnet receives simplified, accurate, and steady learning objectives. Without any modification to the architecture, we deliver a general framework Scala which could be easily built on existing methods.

### 4.1   Framework

Our goal is to build a general framework that makes a ViT $F(\cdot)$ slimmable, i.e., the delivered network can be transformed into different small variants for flexible inference. First, we introduce a hyperparameter $r$ to denote the width ratio of the sub-network $F^r(\cdot)$. Based on our analysis in Fig. 3, ViTs have minimal interpolation ability, which suggests that all subnets have to be individually optimized to achieve decent performance. Following the sandwich rule [37], we sample the smallest $s$, largest $l$ ($l = 1$), and 2 random intermediate width ratios $m_1$, $m_2$ at each iteration during training. The corresponding sub-networks are: $F^s(\cdot)$, $F^l(\cdot)$, $F^{m_1}(\cdot)$ and $F^{m_2}(\cdot)$. and we accumulate the gradients of those subnets at each iteration. At the inference stage, the network $F(\cdot)$ is evaluated at an arbitrary width ratio that has been optimized during training by adjusting $r$.

### 4.2   Isolated Activation

Illustrated in Sec. 3, constant activation of the smallest sub-network $F^s(\cdot)$ ensures its own accuracy at the cost of other subnets' performance. This is a dilemma as there is a significant accuracy drop of $F^s(\cdot)$ if we do not constantly activate it (see Fig. 4), otherwise, the performance of other sub-networks are severely limited. To alleviate this issue, we propose Isolated Activation to disentangle

the representation of $F^s(\cdot)$ from other sub-networks while still constantly activating it. It not only ensures the performance of the lower bound but facilitates the optimization of other subnets as well.

Formally, given the learnable weight $\theta \in \mathbb{R}^{C_o \times C_i \times H \times W}$ of a random layer in ViT ($C_o, C_i$ stands for the output, input channel number, $H, W$ represents the height and width of the convolution kernel, $H = W = 1$ for fully connected layers), the weight of $F^r(\cdot)$ where $r \neq s$ is selected as:

$$\theta^r = \theta\left[:(r \times C_o),\ :(r \times C_i)\right]. \tag{2}$$

In contrast, the smallest sub-network $F^s(\cdot)$ is activated as:

$$\theta^s = \theta\left[-(s \times C_o):,\ -(s \times C_i):\right], \tag{3}$$

where we slice the weights in a reverse direction so that we disentangle the representation of $F^s(\cdot)$ from other sub-networks. With this simple but critical design, we not only ensure the performance of $F^s(\cdot)$ with constant activation, but also alleviate the negative effects it brings.

### 4.3  Scale Coordination

We present the training strategy of Scala in this section. We follow the setting of DeiT [29] to train the full model $F^l(\cdot)$ with knowledge distillation and introduce a distillation token for knowledge transfer between sub-networks. As our goal is to scale down a given network $F(\cdot)$ to multiple smaller variants, we simply choose the pre-trained model itself $F^\circ(\cdot)$ as the external teacher for the full network $F^l(\cdot)$ to facilitate training. To optimize the sub-networks at different scales, we present the Scale Coordination training strategy, which is composed of three techniques: Progressive Knowledge Transfer, Stable Sampling, and Noisy Calibration, to ensure that each subnet receives simplified, accurate, and steady learning objectives.

**Progressive Knowledge Transfer.** Given an input image $v$, the activated sub-network $F^r(\cdot)$ produces two predictions:

$$p^r_{cls},\ p^r_{dis} = F^r(v; \theta^r), \tag{4}$$

where $p^r_{cls}$ and $p^r_{dis}$ denote the prediction generated by the classification and distillation head, respectively. As we activate multiple sub-networks: $F^s(\cdot)$, $F^l(\cdot)$, $F^{m_1}(\cdot)$ and $F^{m_2}(\cdot)$ at each iteration during training, our idea is to utilize the predictions of the larger network to facilitate the optimization of smaller subnets.

Given the sorted width ratio list $R = [s, m_1, m_2, l]$, we utilize the KL divergence [19] loss to progressively distill the knowledge of the larger network into the smaller one:

$$\mathcal{L}^r_{KL} = -\sum_{k=1}^{K} p^{r'}_{dis} \log\left(\frac{p^r_{dis}}{p^{r'}_{dis}}\right), \tag{5}$$

where $K$ represents the number of classes, $r' = R\left[\varsigma(r) + 1\right]$ and $\varsigma(r)$ denotes the index of $r$ in $R$. Instead of using $p^l_{dis}$ as the optimization target for all smaller networks, we ensure each subnet receives simplified learning objective as small subnet have large capacity gap compared to $F^l(\cdot)$ (e.g., $F^l(\cdot)$ is almost 16 times larger than $F^s(\cdot)$ if $s = 0.25$) and minimizing their KL loss complicates the optimization process and leads to inferior performance. With Progressive Knowledge Transfer, we simplify the optimization objective for small sub-networks by utilizing $F^{m_1}(\cdot)$ and $F^{m_2}(\cdot)$ as the teacher assistants to fill the gap between $F^l(\cdot)$ and $F^s(\cdot)$ and train them in a one-shot manner.

**Stable Sampling.** As the knowledge is gradually transferred from the larger network to the smaller one, the two intermediate networks serve as the bridge to connect $F^l(\cdot)$ and $F^s(\cdot)$, as $F^{m_2}(\cdot)$ is the student of $F^l(\cdot)$ and $F^{m_1}(\cdot)$ is the teacher of $F^s(\cdot)$. Therefore, we need to carefully control the width ratios $m_1$ and $m_2$ to prevent the obvious model capacity variation.

Concretely, we introduce the slicing granularity $\epsilon$ and the number of networks $X$ we can deliver (including the full model) with a single ViT is denoted as:

$$X = \frac{(l - s)}{\epsilon} + 1. \tag{6}$$

Then, we divide the slicing bound $B = \frac{(s, l)}{\epsilon}$ into two smaller ones:

$$B_1 = \frac{(s, \bar{m}]}{\epsilon},\ \ B_2 = \frac{(\bar{m}, l)}{\epsilon}, \tag{7}$$

where $\bar{m} = \frac{(s+l)}{2}$ and $\tilde{m}_1$, $\tilde{m}_2$ will be the random integer sampled from the uniform distribution $B_1$, $B_2$, respectively. Thus, $m_1$ and $m_2$ are defined as:

$$m_1 = \tilde{m}_1 \times \epsilon, \quad m_2 = \tilde{m}_2 \times \epsilon, \tag{8}$$

and we ensure the model capacity gap between the four networks is stable and secure the learning objective for each subnet is steady.

**Noise Calibration.** Although all the subnets receive guidance from larger networks, a notable issue is that the predictions from the teacher are not always accurate, sometimes even noisy, especially at the early training stage. To avoid the noisy signal dominating the optimization direction, we first calculate the Cross-Entropy loss by:

$$\mathcal{L}^r_{CE} = -\sum_{k=1}^{K} \hat{y}_k \log \left( p^r_{cls} \right), \tag{9}$$

where $\hat{y}_k$ represents the one-hot label for class $k$. Then, we calibrate the noise by combining the KL divergence loss and Cross-Entropy loss:

$$\mathcal{L}^r = \mathcal{L}^r_{CE} + \lambda \cdot \mathcal{L}^r_{KL}, \tag{10}$$

where $\lambda$ is a hyperparameter used to balance the two losses and we empirically let $\lambda = 1$ in our implementations. By doing so, we mitigate the negative effects brought by the noisy predictions of the teacher model and ensure each subnet is guided by the accurate learning objectives.

## 5 Experiments

We validate Scala with the plain ViT structure DeiT [29]. We first analyze of the main property of Scala and compare our method with the state-of-the-art method SN-Net [25] and Separate Training (ST) at a larger scale. Moreover, we examine the transferability of Scala and its application on Semantic Segmentation. Finally, we provide ablations to validate the efficacy of our designs.

### 5.1 Experiment Settings

All the object recognition experiments are carried out on ImageNet-1K [8]. We follow the training recipe of DeiT [29] and conduct the experiments on 4 V100 GPUs. For Scala, we set $s = 0.25$, $l = 1.0$, and $\epsilon = 0.0625$ so that we could enable a single ViT to represent 13 different networks ($X = 13$) with a large slicing bound (i.e., $F^l(\cdot)$ is almost 16 times larger than $F^s(\cdot)$).

Table 1: Comparison with scaling baseline methods AutoFormer [4], US-Net [37] and Separate Training under different width ratios $r$. The best results are bold-faced.

| Method | Param | $r = 0.25$ | | $r = 0.50$ | | $r = 0.75$ | | $r = 1.00$ | |
|---|---|---|---|---|---|---|---|---|---|
| | | Acc1. | GFLOPs | Acc1. | GFLOPs | Acc1. | GFLOPs | Acc1. | GFLOPs |
| AutoFormer [4] | 22M | 50.8% | 0.4 | 65.6% | 1.3 | 69.5% | 2.7 | 69.8% | 4.6 |
| US-Net [37] | 22M | 52.7% | 0.4 | 60.1% | 1.3 | 66.9% | 2.7 | 73.2% | 4.6 |
| Separate Training | 43M | 45.8% | 0.4 | 65.1% | 1.3 | 70.7% | 2.7 | 75.0% | 4.6 |
| Scala | 22M | **58.7**% | 0.4 | **68.3**% | 1.3 | **73.3**% | 2.7 | **76.1**% | 4.6 |

### 5.2 Proof-of-Concept

In this part, we conduct experiments over DeiT-S [29] for 100-epoch training to prove the concept.

**Comparison with scaling baselines.** We compare Scala with multiple scaling baselines, including: (1) AutoFormer [4]: we apply this ViT-based supernet training method into our setting to scale through width; (2) US-Net [37]: the prior work that obtains similar performance with ST over the CNN structure; (3) Separate Training (ST): we repetitively train the model with different widths from scratch and evaluate them individually. Tab. 1 shows that AutoFormer lags behind Scala remarkably as we target to scale in a wider range. US-Net shows significantly worse performance compared to ST which indicates that scaling down ViT is a more challenging problem compared to the CNN architecture. Nevertheless, Scala achieves better performance compared to ST at all width ratios with one-shot training, reducing the storage costs of saving multiple models observably.

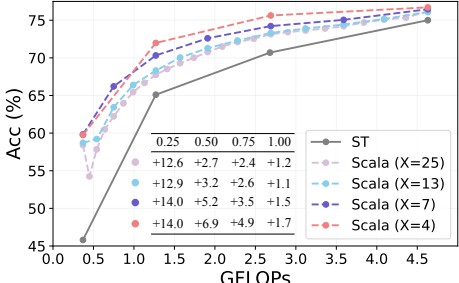

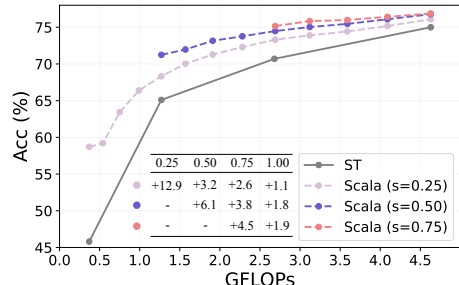

Figure 5: Comparisons of Scala with different slicing granularity and Separate Training (ST). Improvements over ST are shown.

Figure 6: Comparisons of Scala with different slicing bounds and Separate Training (ST). Improvements over ST are shown.

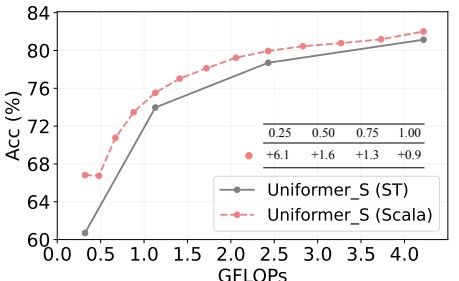

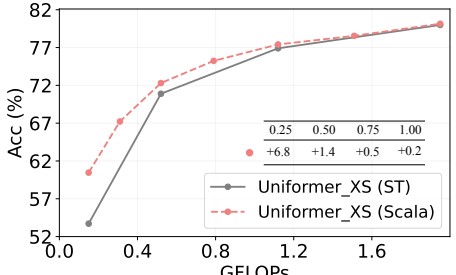

Figure 7: Comparisons of Scala and Separate Training (ST) over the CNN-ViT hybrid architecture Uniformer-S [20]. Improvements over ST are shown.

Figure 8: Comparisons of Scala and Separate Training (ST) over lightweight model Uniformer-XS [20] with token pruning. Improvements over ST are shown.

**Slicing Granularity and Bound.** Fig. 5 shows the results of various slicing granularity $\epsilon$. First, Scala outperforms ST with different $\epsilon$ and the advantage at small width ratios is more obvious, which promises its application on edge devices. Moreover, it is shown that less fine-grained granularity $\epsilon$ results in better overall performance with the same slicing bound as the expected training epochs $\xi$ for intermediate subnets increase correspondingly. We further conduct experiments with different $s$ while fixing $l$ and $\epsilon$ to study the effect of the slicing bound. Fig. 6 shows that smaller bounds lead to markedly better performance and it further verifies that slicing through a large bound is intrinsically more difficult, which distinguishes Scala from the supernet training methods [2, 38, 4] in NAS.

**Application on Hybrid Structures.** We experiment Scala on the CNN-ViT hybrid architecture Uniformer-S [20]. As Uniformer contains Batch Normalization (BN) [15] which cannot be directly evaluated after slicing because of normalization shifting [39], we calibrate the statistics of BN before inference following [37]. Shown in Fig. 7, Uniformer-S is scaled down to 13 different variants with better performance compared to ST, demonstrating the generalization ability of Scala. However, performing BN calibration at each width ratio requires considerable extra effort. This highlights the benefit of ViT, as Layer Normalization (LN) allows direct evaluation without additional operations.

**Application on Lightweight Structures.** We further validate Scala on lightweight structure Uniformer-XS [20] which integrates the design of token pruning and train these methods for 150 epochs. Shown in Fig. 8, Scala still matches the performance of ST and exhibits a significant advantage at small width ratios, which promises its application on edge devices with a limited budget.

**Fast Interpolation of Slimmable Representation.** Training models with different slicing granularity $\epsilon$ from scratch is time-consuming and here we show that the slimmable representation of certain granularity can be scaled to others with a small amount of training epochs. Specifically, we train the model with the original $\epsilon$ for 70 epochs and decrease the value of $\epsilon$ in the last 30 epochs to deliver more sub-networks for higher inference flexibility. Fig. 9 shows the results of fast interpolation are similar to those trained from scratch and the newly appeared sub-networks are quickly interpolated to achieve decent performance. We further increase $\epsilon$ for sub-networks with higher performance and the phenomenon shown in Fig. 10 is similar to down interpolation. Besides, we observe that the accuracy of abandoned sub-networks gradually decreases but they maintain the performance to a great extent.

**Slimmable Ability across Architectures.** We examine the slimmable ability of different architectures in Tab. 2 by applying Scala on different architectures and evaluating these networks at unseen width

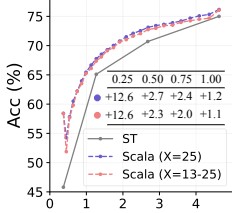 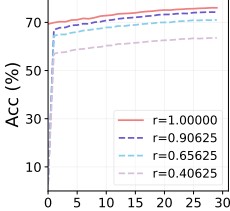 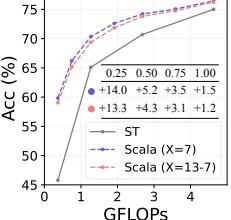 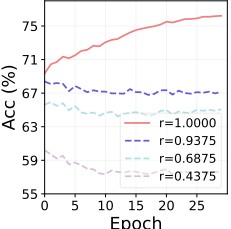

(a) Results of w/ and w/o down interpolation.

(b) Accuracy change of the introduced subnets.

(a) Results of w/ and w/o up interpolation.

(b) Accuracy change of the abandoned subnets.

Figure 9: Down Interpolation. We reduce the slicing granularity in the last 30-epoch training of DeiT-S [29] on ImageNet-1K to deliver more sub-networks for higher inference flexibility.

Figure 10: Up Interpolation. We increase the slicing granularity in the last 30-epoch training of DeiT-S [29] on ImageNet-1K to deliver fewer sub-networks for higher performance.

Table 2: Slimmable ability examination over different architectures on ImageNet-1K under various width ratios. The interpolated results are shown in blue color.

| Architecture | Model | Top-1 Acc. (%) | | | | | | |
|---|---|---|---|---|---|---|---|---|
| | | 0.40625 | 0.4375 | 0.46875 | 0.50 | 0.53125 | 0.5625 | 0.59375 |
| ViT | DeiT-S | 1.7 | 66.4 | 1.6 | 68.3 | 1.9 | 70.0 | 1.9 |
| CNN-ViT | Uniformer-S | 30.4 | 73.5 | 27.8 | 75.5 | 32.6 | 77.0 | 37.5 |
| CNN | MobileNet v2 | 58.8 | 60.4 | 60.7 | 61.6 | 61.8 | 64.3 | 64.4 |

ratios to explore the interpolation ability. CNN exhibits very strong interpolation ability as the performance at unseen widths lies in the range of trained width ratios. In contrast, CNN-ViT and ViT suffer from remarkable performance decreases to different extents and ViT achieves almost zero accuracy which further validates that the problem we target to solve, i.e., slicing ViT, is the most challenging one.

## 5.3 Comparisons over Extended Training

In this section, we perform training over DeiT-B [29] for 300-epoch training following the standard protocol on ImageNet-1K [8] to compare with the state-of-the-art.

**Comparisons with state-of-the-art.** SN-Net [25] is state-of-the-art work that supports flexible inference on ViT. Specifically, it utilizes several pre-trained models (e.g., DeiT-Ti/S/B) to construct a supernet and inserts additional layers to build dynamic routes for flexible inference. Shown in Tab. 3, we empirically compare Scala with SN-Net over DeiT-B following the standard 300-epoch training protocol [29]. Scala obtains similar performance with SN-Net at large width ratios and clearly outperforms it at small computational budgets. Besides, SN-Net has to preserve the parameters of multiple models and additional layers, while Scala only needs to keep the weights of the full network. When adopting the stronger teacher network [27] as SN-Net does, Scala outperforms SN-Net with an average improvement of 1.6% across all width ratios.

**Comparisons with Separate Training.** In Tab. 4, we compare with ST on DeiT-B [29] with longer training process, i.e., 300-epoch training, where $r = 0.25, 0.50, 1.00$ corresponds to DeiT-Ti, DeiT-S and DeiT-B, respectively. Scala exhibits a clear advantage at $r = 0.25$ and matches the performance of ST except $r = 0.50$ due to significantly less training time. When $X = 7$, we can achieve similar performance at $r = 0.50$ with 40% training epochs of ST. Further reducing $X$ to 4, resulting in the constant activation of the two intermediate networks, allows us to consistently outperform ST at all width ratios. This substantiates the effectiveness of Scala and the slimmable representation.

Table 3: Comparison with SN-Net [25] over DeiT-B [29] on ImageNet-1K. ◇, ♣ denotes utilizing DeiT-B [29], RegNetY-16GF [27] as the teacher model to facilitate training.

| Method | Param | Top-1 Acc. (%) | | | | | | |
|---|---|---|---|---|---|---|---|---|
| | | 0.25 | 0.375 | 0.50 | 0.625 | 0.75 | 0.875 | 1.00 |
| ♣ SN-Net [25] | 118M | 70.6 | 74.8 | 79.5 | 79.3 | 81.2 | 81.9 | 81.9 |
| ◇ Scala | 86M | 75.3 | 76.9 | 79.3 | 80.5 | 81.2 | 81.6 | 82.0 |
| ♣ Scala | 86M | **75.4** | **77.2** | **79.7** | **80.9** | **81.8** | **82.2** | **82.9** |

Table 4: Comparison with Separate Training (ST) over DeiT-B [29] on ImageNet-1K under different width ratios $r$. $\xi$ denotes the expected training epochs of each model.

| Method | Param | $r = 0.25$ | $r = 0.50$ | | $r = 0.75$ | | $r = 1.00$ |
|---|---|---|---|---|---|---|---|
| | | Acc1. | Acc1. | $\xi$ | Acc1. | $\xi$ | Acc1. |
| ST | 163M | 72.2% | 79.9% | 300 | 81.0% | 300 | 81.8% |
| Scala (X=13) | 86M | 75.3% | 79.3% | 55 | 81.2% | 55 | 82.0% |
| Scala (X=7) | 86M | 75.3% | 79.7% | 120 | 81.4% | 120 | 82.0% |
| Scala (X=4) | 86M | **75.6%** | **80.9%** | 300 | **81.9%** | 300 | **82.2%** |

## 5.4 Transferability

To assess the transferability of Scala, we employ DeiT-B [29] as the backbone for a 300-epoch pre-training on ImageNet-1K and leverage the foundation model DINOv2-B [24] as the teacher network to inherit good behaviors. Our study aims to address two key questions:

**Whether the slimmable representation can be transferred to downstream tasks?** As depicted in Fig. 11a, Scala consistently outperforms Separate Training (ST) across all width ratios, despite the intermediate sub-networks being trained for approximately 55 epochs. After that, we conduct linear probing on video recognition dataset UCF101 with 8 evenly sampled frames and average their features for the final prediction. For the classification head added on Scala, we make it slimmable to fit the features with various dimensions and follow the same training protocol as in object recognition. In Fig. 11b, two notable observations emerge: (1) Scala consistently outperforms ST across different width ratios on the UCF101 dataset, implying the great transferability of the slimmable representation; (2) Scala retains its slimmable ability when applied to a new task and exhibits promising performance across a wide slicing range (10~141 GFLOPs), promising its application on other downstream tasks.

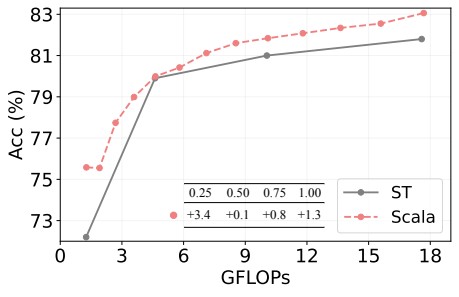
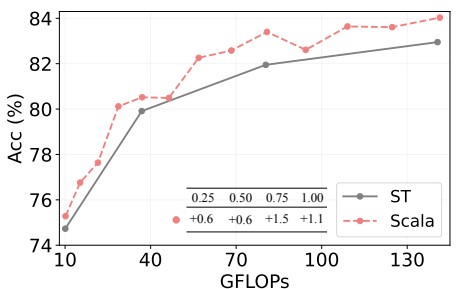

(a) Pre-training on ImageNet-1K.      (b) Linear probing on UCF101 with 8 sampled frames.

Figure 11: Transferability of Scala. We first conduct pre-training on ImageNet-1K with the help of foundation model DINOv2-B [24]. Then we conduct linear probing on video recognition dataset UCF101. Improvements over ST are shown.

**Whether the generalization ability can be maintained in the slimmable representation?** Inspired by the work [43] which replicates the success of vision foundation models on ImageNet-1K, we remove all the Cross-Entropy losses during training to alleviate the dataset bias issue and inherit the strong generalization ability of the teacher network DINOv2. Then we conduct linear probing on 12 fine-grained classification datasets following the setup in DINOv2. Tab. 5 shows that Scala significantly outperforms DeiT variants on the average performance of fine-grained classification which suggests that Scala indeed inherits the fruitful knowledge from DINOv2 with remarkable improvement in its generalization ability. Moreover, the improvement over DeiT does not decrease when we scale down the width ratios during inference and it indicates that Scala maintains the flexible inference capability very well even though it contains more knowledge than before.

Table 5: Comparison of Scala and DeiT on 12 fine-grained classification datasets. Scala-B is distilled from DINOv2-B on ImageNet-1K for 300 epochs and then we conduct linear probing on fine-grained datasets. The average accuracy of 12 datasets is shown in the last column.

| Method | $r$ | Arch | Aircraft | Cal101 | Cars | C10 | C100 | DTD | Flowers | Food | Pets | SUN | VOC | CUB | Average |
|---|---|---|---|---|---|---|---|---|---|---|---|---|---|---|---|
| DeiT-B | 1.00 | ViT-B | 49.1 | 90.7 | 57.1 | 95.6 | 81.1 | 70.3 | 90.7 | 76.9 | 92.8 | 63.0 | 83.0 | 72.8 | 77.0 |
| DeiT-S | 1.00 | ViT-S | 44.5 | 90.3 | 49.4 | 94.1 | 77.7 | 69.5 | 88.8 | 72.1 | 92.1 | 60.5 | 83.3 | 71.9 | 74.5 |
| DeiT-Ti | 1.00 | ViT-Ti | 39.9 | 88.8 | 39.7 | 90.6 | 72.5 | 67.0 | 85.1 | 65.0 | 91.1 | 56.1 | 81.6 | 69.4 | 70.6 |
| Scala-B | 1.00 | ViT-B | 63.0 | 95.1 | 74.3 | 97.3 | 86.6 | 78.0 | 97.3 | 83.4 | 94.6 | 69.7 | 87.6 | 84.1 | **84.2** |
| Scala-B | 0.50 | ViT-S | 55.2 | 93.9 | 64.4 | 95.5 | 81.9 | 74.9 | 96.0 | 78.6 | 94.0 | 65.9 | 86.1 | 81.1 | **80.6** |
| Scala-B | 0.25 | ViT-Ti | 49.6 | 92.1 | 55.4 | 93.6 | 78.1 | 72.1 | 93.4 | 72.8 | 92.9 | 61.4 | 84.2 | 76.7 | **76.8** |

## 5.5 Dense Prediction

In previous sections, we have validated the effectiveness of Scala on classification tasks, we further examine whether the slimmable representation could be transferred for dense prediction task like

Table 6: Evaluation of slimmable representation on dense prediction task Semantic Segmentation over ADE20K [45] dataset. We equipped the pre-trained Uniformer-S [20] from Fig. 7 with Semantic FPN [16] and compare the sub-networks extracted from Scala with Separate Training (ST).

| Backbone | mIoU. (%) | | | |
|---|---|---|---|---|
| | 0.25 | 0.50 | 0.75 | 1.00 |
| Uniformer-S (ST) | 33.9 | 40.4 | 42.6 | 45.3 |
| Uniformer-S (Scala) | **35.0** | **40.7** | **43.7** | **46.1** |

Table 7: Ablation study of Scala over DeiT-S [29] on ImageNet-1K under various width ratios. IA, PKT, SS, NC denote Isolated Activation, Progressive Knowledge Transfer, Stable Sampling, Noise Calibration, respectively. The best results are bold-faced.

| Method | Top-1 Acc. (%) | | | | | | |
|---|---|---|---|---|---|---|---|
| | 0.25 | 0.375 | 0.50 | 0.625 | 0.75 | 0.875 | 1.00 |
| Scala | **58.7** | **63.4** | **68.3** | **71.3** | **73.3** | **74.4** | 76.1 |
| w/o IA | 57.3 | 61.5 | 66.4 | 69.8 | 72.0 | 73.4 | 75.8 |
| w/o PKT | 53.0 | 60.1 | 65.6 | 68.8 | 71.7 | 73.7 | 76.2 |
| w/o SS | 58.7 | 62.7 | 68.1 | 71.3 | 73.1 | 74.2 | 75.9 |
| w/o NC | 50.2 | 62.7 | 67.2 | 70.5 | 72.7 | 74.0 | **76.3** |

semantic segmentation. We utilize the pre-trained model Uniformer-S [20] drawn from Fig. 7, which has a hierarchical design and is obtained by 100-epoch training (our results lag behind official results where the backbone is trained for 300 epochs), and equip it with Semantic FPN [16]. To compare with Separate Training (ST), we extract four subnets from Scala (Uniformer-S) and train them separately. Shown in Tab. 6, Scala outperforms ST at all widths which verifies the slimmable representation benefits the downstream tasks. Note that we do not scale the decoder as it involves extra designs and is out of the scope of this work. However, we show that the slimmable representation can be generalized to semantic segmentation as feature extractors because the feature maps are spatially intact, promising its application as an end-to-end slimmable framework on dense prediction tasks.

## 5.6 Ablation Study

We conduct ablation to examine the effectiveness of our designs in Tab. 7. First, we build Scala without Isolated Activation so that the smallest sub-network will entangle with others and it shows an obvious performance drop at all width ratios. Then, we remove Progressive Knowledge Transfer (PKT) and pass the knowledge from $F^l(\cdot)$ to smaller subnets through classification token following US-Net [37]. It shows much worse performance, especially at small ratios, which proves the strength of PKT as it implicitly introduces some teacher assistants to simplify the optimization objective for small sub-networks. Further, we random sample the width ratios of $m_1$ and $m_2$ between $(s, l)$ and compare it with Stable Sampling (SS). The results are slightly inferior to SS which suggests SS is helpful in securing the steady learning objective for each sub-network. Finally, we remove Noise Calibration (NC) from Scala and only use the predictions from larger networks to guide the small subnets. It shows remarkable performance drops at small width ratios, where the noise from the teacher network is most obvious, demonstrating the effectiveness of NC in calibrating the noise and providing accurate signals for sub-networks.

## 6 Conclusion and Limitations

In this paper, we observed that smaller ViTs are intrinsically the sub-networks of a large ViT with different width ratios. However, slicing ViT is very challenging due to its poor interpolation ability. To address this issue, we proposed Scala to enable a single network to represent multiple smaller variants with flexible inference capability. Specifically, we proposed Isolated Activation to disentangle the representation of the smallest subnet from others and presented Scale Coordination to ensure the sub-network receives simplified, steady, and accurate learning objectives. Extensive experiments on different tasks prove that Scala, requiring only one-shot training, outperforms the state-of-the-art method under different computations and matches the performance of Separate Training with significantly fewer parameters, promising the potential as a new training paradigm.

One limitation of Scala is the longer training time compared to conventional supervised learning of a single model, attributable to the activation of multiple subnets during training. Nevertheless, our training time is obviously less than separately training all the sub-networks. In the future, we aim to enhance the training efficiency of Scala.

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

# A Appendix

## A.1 Implementation Details

For Separate Training (ST), we follow the exact training strategy of the official DeiT [29] and Uniformer [20] setting. We use random horizontal flipping, random erasing [44], Mixup [42], CutMix [40], and RandAugment [6] for data augmentation. AdamW [23] is utilized as the optimizer with a momentum of 0.9 and a weight decay of 0.05. We set the learning rate to 1e-3 and decay with a cosine shape. The models are trained on 4 V100 and 8 A100 GPUs with a total batch size of 1024. We adopt Exponential Moving Average (EMA) following the official setting.

While we utilize the pre-trained ST model ($r = 1.00$) as the teacher for $F^l (\cdot)$ to facilitate training as mentioned in the Scale Coordination section in the main paper, we adopt a larger learning rate (2e-3) and mild data augmentation (reduce the magnitude for RandAugment [6] to 1 and turn off repeated augmentation) because Scale Coordination already regularizes the network training strongly. At every training iteration, we activate four sub-networks separately based on Stable Sampling and accumulate their gradients for backpropagation. The rest hyperparameters are set as the same as those in Separate Training.

## A.2 Slimmable Ability of Vanilla Representation

Tab. 1 shows that ViT is not slimmable if we directly evaluate the vanilla pre-trained model at other widths. Here, we further explore the slimmable ability of vanilla representation and fine-tune the vanilla pre-trained model with Scala. Tab. 8 shows that fine-tuning obtains obviously worse performance at small width ratios compared to training from scratch, which denotes that the vanilla representation is not slimmable and is essentially different from the slimmable representation.

Table 8: Comparisons of training from scratch and fine-tuning on ImageNet-1K.

| Method | Top-1 Acc. (%) | | | |
|---|---|---|---|---|
| | 0.25 | 0.50 | 0.75 | 1.00 |
| DeiT-S [29] | - | - | - | 75.0 |
| + Fine-tune | 23.4 | 52.3 | 65.4 | 74.0 |
| + Scratch | **58.7** | **68.3** | **73.3** | **76.1** |

## A.3 Larger Slicing Bound

As discussed in the main text, the slicing bound has a huge impact on the performance of Scala. Here we further expand the slicing bound from $[0.25, 1.00]$ to $[0.125, 1.000]$. As shown in Fig. 12, Scala suffers from an obvious performance drop at $r = 0.25$ as it is not constantly activated in the new setting. Nevertheless, our method still manages to outperform ST at all width ratios and shows a significant advantage at the smallest ratio $r = 0.125$.

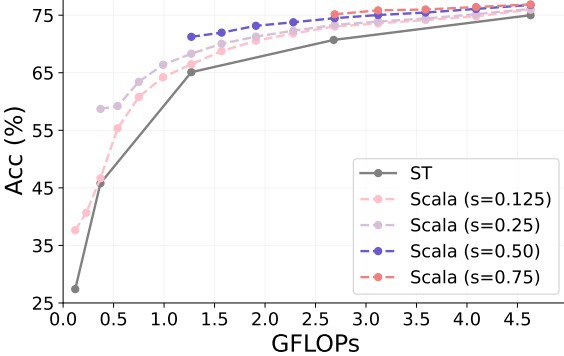

Figure 12: Increase the slicing bound to $[0.125, 1.000]$ over DeiT-S [29] on ImageNet-1K.

## A.4 Longer Training Process

Previous experiments validate that Scala achieves achieves performance comparable to that of Separate Training (ST) on DeiT-B [29] in the 300-epoch training setting, even though the intermediate sub-

Table 9: Comparison with Separate Training (ST) over DeiT-B [29] on ImageNet-1K with different training epochs under different $r$. $\xi$ denotes the expected training epochs of each model.

| Method | Epoch | $r = 0.25$ | $r = 0.50$ | | $r = 0.75$ | | $r = 1.00$ |
|---|---|---|---|---|---|---|---|
| | | Acc1. | Acc1. | $\xi$ | Acc1. | $\xi$ | Acc1. |
| ST | 300 | 72.2% | 79.9% | 300 | 81.0% | 300 | 81.8% |
| Scala (X=13) | 300 | 75.3% | 79.3% | 55 | 81.2% | 55 | 82.0% |
| Scala (X=7) | 300 | 75.3% | 79.7% | 120 | 81.4% | 120 | 82.0% |
| Scala (X=13) | 400 | 75.7% | 79.5% | 73 | 81.5% | 73 | 82.3% |
| Scala (X=7) | 400 | 76.0% | 80.1% | 160 | 81.7% | 160 | 82.4% |

networks training time is much less. We further extend the training process to 400 epochs in this section and the results are shown in Tab. 9. The overall performance at various width ratios is improved with longer training and Scala ($X = 7$) outperforms ST at all widths even though the expected training epochs for intermediate sub-networks are still much less than ST.

## A.5 More Ablation Studies on Activation Method

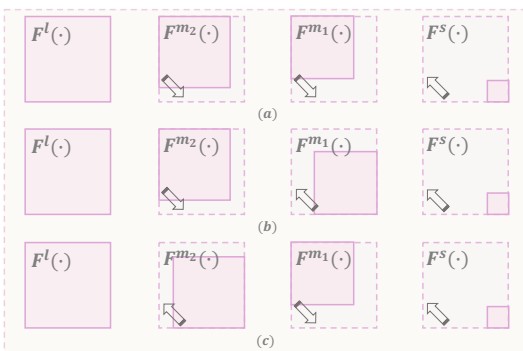

Figure 13: Illustration of different activation methods.

Table 10: Ablation study of various activation methods over DeiT-S [29] on ImageNet-1K under various widths. Different activation methods are illustrated in Fig. 13. The best results are bold-faced.

| Method | Top-1 Acc. (%) | | | | | | |
|---|---|---|---|---|---|---|---|
| | 0.25 | 0.375 | 0.50 | 0.625 | 0.75 | 0.875 | 1.00 |
| Scala+(a) | **58.7** | **63.4** | **68.3** | 71.3 | 73.3 | 74.4 | **76.1** |
| Scala+(b) | 57.6 | 61.3 | 64.9 | **72.4** | **74.1** | **74.7** | **76.1** |
| Scala+(c) | 58.3 | 63.3 | 67.2 | 69.4 | 72.0 | 72.9 | 74.8 |

We validated the effectiveness of Isolated Activation by removing this component and we further conduct more ablation studies on the activation methods where the designs are illustrated in Fig. 13. As shown in Tab. 10, choice (b) leads to slightly better performance at $F^{m_2}(\cdot)$, but the performance drops at $F^{m_1}(\cdot)$ significantly as it is entangled with $F^s(\cdot)$ and the over-emphasize of $F^s(\cdot)$ adversely affect its performance. On the other hand, choice (c) results in a similar performance at $F^{m_1}(\cdot)$, but the accuracy decreases significantly at $F^{m_2}(\cdot)$ which further verifies our hypothesis that $F^s(\cdot)$ should be isolated to reduce its negative impact on other sub-networks.

## A.6 Verification of Slimmable Ability

In the main text, we found that ViT has the minimal interpolation ability compared to the CNN structure. This suggests that optimizing larger sub-networks does not directly contribute to the performance improvement of smaller variants, even though their weights are shared in a nested nature. A further question is, whether ViT can still maintain the slimmable ability for the unseen width ratios during training.

To verify this point, we respectively fix the width ratios of $m_2, m_1$ to 0.8125 and 0.4375 during training, so that only one sub-network is optimized at each range. Fig. 14 shows that the accuracy of unseen sub-networks is very low due to the lack of interpolation ability. Nevertheless, the performance

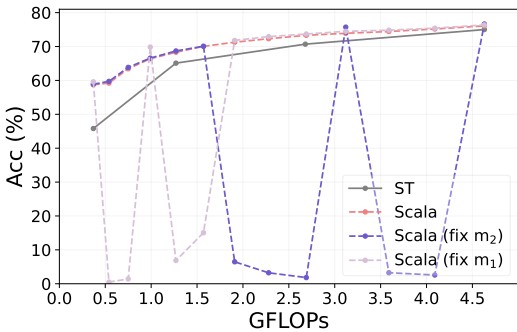

Figure 14: Verification of Slimmable Ability over DeiT-S [29] on ImageNet-1K. We respectively fix the width ratio of $m_2, m_1$ to 0.8125 and 0.4375, and observe the performance at other ratios is not affected.

at other width ratios remains similar to the default setting even though their weights are shared with each other. This indicates that the correlation between sub-networks in ViT is weak and further highlights how challenging this problem is.

## A.7 Comparisons with Distillation Baselines

Table 11: Comparison with baseline methods US-Net and Separate Training (ST) by adding an external teacher on ImageNet-1K dataset under different width ratios $r$. All methods are built upon DeiT-S [29] which are trained for 100 epochs and $\xi$ denotes the expected training epochs of each model. The best results are bold-faced.

| Method | Param | $r = 0.25$ | | | $r = 0.50$ | | | $r = 0.75$ | | | $r = 1.00$ | | |
|---|---|---|---|---|---|---|---|---|---|---|---|---|---|
| | | Acc1. | $\xi$ | GFLOPs | Acc1. | $\xi$ | GFLOPs | Acc1. | $\xi$ | GFLOPs | Acc1. | $\xi$ | GFLOPs |
| US-Net | 22M | 52.7% | 100 | 0.4 | 60.1% | 18 | 1.3 | 66.9% | 18 | 2.7 | 73.2% | 100 | 4.6 |
| ST | 43M | 45.8% | 100 | 0.4 | 65.1% | 100 | 1.3 | 70.7% | 100 | 2.7 | 75.0% | 100 | 4.6 |
| US-Net+KD | 22M | 51.6% | 100 | 0.4 | 58.7% | 18 | 1.3 | 66.6% | 18 | 2.7 | 73.8% | 100 | 4.6 |
| ST+KD | 43M | 48.8% | 100 | 0.4 | 67.5% | 100 | 1.3 | **73.3**% | 100 | 2.7 | **76.4**% | 100 | 4.6 |
| Scala | 22M | **58.7**% | 100 | 0.4 | **68.3**% | 18 | 1.3 | **73.3**% | 18 | 2.7 | 76.1% | 100 | 4.6 |

While we have shown that Scala outperforms baseline methods US-Net [37] and Separate Training (ST), we further compare Scala with much stronger baselines, by adding an external teacher to their full network during training. Specifically, we adopt the pre-trained full model from ST as the teacher and conduct knowledge distillation for models with different widths separately. Tab. 11 shows that ST+KD exhibits similar performance at larger width ratios with Scala, despite that Scala clearly outperforms ST+KD at smaller widths, promising its application on edge devices. Although obtaining a better full model, US-Net+KD exhibits worse performance at smaller width ratios because it utilizes the full network as the teacher for all subnets and this phenomenon verifies our motivation of proposing Progressive Knowledge Transfer.

## A.8 Comparisons in Training Time

Assuming to deliver 13 models in the end, we compare the training time (100 Epoch) of Scala with US-Net [37], Separate Training on 8 A100 GPUs. The difference between US-Net and Scala is not large as the transformer architecture has been well-optimized on GPU and we do observe a significant time gap between Scala and Separate Training as they have to train 13 models iteratively. Moreover, Scala can be configured to deliver 25 models without an increase in training time as we sample 4 networks at each iteration in all scenarios which further highlights our strengths.

Table 12: Comparisons of training time with baseline methods on 8 A100 GPUs.

| Method | Training Hours |
|---|---|
| Separate Training | 123 |
| US-Net | 20 |
| Scala | 21 |

## A.9 Comparisons with MatFormer

MatFormer [18] only slices the FFN block in the transformer architecture so it offers a minor computational adjustment space and we adapt their method on DeiT-S to compare with Scala. Fig. 15 shows that Scala achieves comparable performance with it (better in most cases) when $s = 0.5$ with a larger adjustment scope.

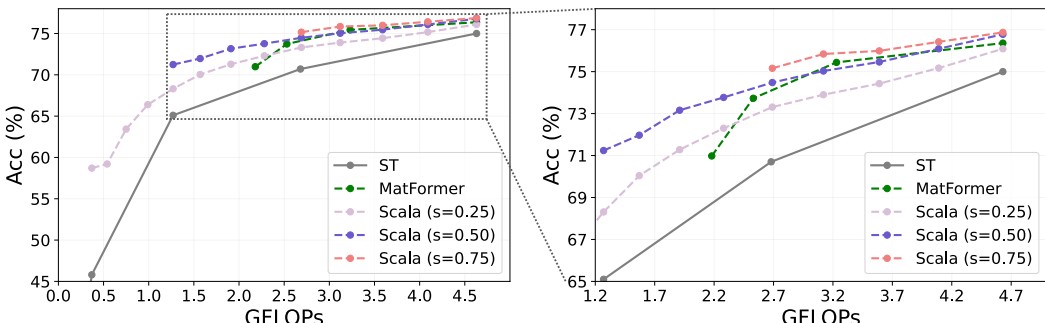

Figure 15: Comparison of Scala and MatFormer over DeiT-S. Scala offers a significantly broader scope for computational adjustment compared to MatFormer as MatFormer only scales the FFN block in ViT. The right figure provides a detailed magnification of the left figure.

