# OpenReview forum: "Slicing Vision Transformer for Flexible Inference"
_NeurIPS.cc/2024/Conference — NeurIPS 2024 poster_

### Official Review · Reviewer_7ef6 · 2024-07-12

**Soundness:** 2
**Presentation:** 2
**Contribution:** 2
**Rating:** 4
**Confidence:** 5

**Summary:**

The paper targets scaling down Vision Transformers (ViT) to fit environments with dynamically changing resource constraints. The authors propose Scala, a framework enabling a single network to represent multiple smaller ViTs with flexible inference capability by activating various subnets during training. Scala introduces Isolated Activation to disentangle the smallest sub-network and uses Scale Coordination to provide stable and accurate learning objectives. Empirical validations on different tasks show that Scala achieves scalable representation with one-shot training, matching the performance of Separate Training without modifying the original ViT structure. Scala demonstrates an average improvement of 1.6% on ImageNet-1K compared to previous methods, using fewer parameters.

**Strengths:**

1. The problem is important in practice.
2. The experimental results seem decent.

**Weaknesses:**

1. My major concern is that, the same aim of adapting ViTs to dynamically changing resource constraints, can also be achieved by multi-exit networks, e.g., [*1, *2, *3]. However, the paper does not discuss these highly relevant works or compare with them. Hence, I vote for rejection.
2. The method seems to lack novelty. 'smaller ViTs are intrinsically the sub-networks of a larger ViT with different widths' is not a surprising observation. The key techniques (e.g., Isolated Activation and Knowledge Distillation) are not new (naive or have been widely adopted).


[*1] Huang, Gao, et al. "Multi-Scale Dense Networks for Resource Efficient Image Classification." International Conference on Learning Representations. 2018.

[*2] Wang, Yulin, et al. "Not all images are worth 16x16 words: Dynamic transformers for efficient image recognition." Advances in neural information processing systems 34 (2021): 11960-11973.

[*3] Han, Yizeng, et al. "Dynamic perceiver for efficient visual recognition." Proceedings of the IEEE/CVF International Conference on Computer Vision. 2023.

**Questions:**

Please refer to Weaknesses.

**Limitations:**

The authors have addressed the limitations and potential negative societal impacts of their work.

---

> ### Author Rebuttal · Authors · 2024-08-07
>
> We appreciate the Reviewer's feedback. We provide further explanations to clarify the Reviewer's concerns based on several key points as below.
>
> ***
>
> **Weakness 1: Discussion with dynamic networks.**
>
> We thank the Reviewer for the valuable suggestion and we will add discussion with those dynamic networks in the final version, including MSDNet[1], RANets[2], GFNet[3], DVT[4], CF-ViT[5], Dyn-Perceiver[6], etc.
>
> However, the differences between our work and this line of research are:
> (1) Motivation: dynamic networks are designed to reduce inference costs with dynamic computational graphs; Scala is proposed to make ViT become slimmable without touching the specific architecture design.
> (2) Method: dynamic networks tailor their computational graphs to accommodate varying inputs, thereby optimizing resource allocation on a per-sample basis. In contrast, Scala treats every sample equally and modulates the computational costs by adjusting the width ratios which aligns with the inherit design of ViT and its inference costs can be explicitly controlled by a single hyperparameter $r$.
>
> Essentially, dynamic networks are agnostic with Scala and we will consider the integration of those two lines of research as our future plan.
>
> **Weakness 2: Lack of novelty.**
>
> Thank you for the comment. While we do agree that 'smaller ViTs are intrinsically the sub-networks of a larger ViT with different widths' is not a surprising observation, making ViTs become slimmable is indeed non-trivial.
>
> According to our analysis in Sec. 2, ViTs display minimal interpolation ability which emphasizes that the problem we target to solve is very challenging as the intermediate subnets ($X=13$) are only trained for around 18 epochs and they are expected to match the performance of networks that are trained for 100 epochs.
>
> To the best of our knowledge, we are the first to propose Isolated Activation, which seems simple but is very effective as we have abundant analysis (Sec. 3) to support it. Moreover, the knowledge distillation in Scala is different from the conventional setting. Instead of choosing the full model as the teacher for all subnets, we enable the intermediate networks to be the teacher as well which fills the model gap between $F^{l}\left (\cdot  \right )$ and $F^{s}\left (\cdot  \right )$ and provide simplified optimization objective for small subnets. This is non-trivial as we find adding a teacher network to the baseline method US-Net will result in performance decreases in the smaller subnets (please see our reply to Reviewer rLHb) which further supports our motivation to propose Progressive Knowledge Transfer. Moreover, Noise Calibration is critical in our setting as well because those intermediate teachers cannot provide faithful predictions at the early stages and it is essential to add Cross-Entropy loss to those subnets during training which is overlooked in previous baseline methods.
>
>
> Based on these factors, Scala can solve this challenging problem in a simple but effective way.
>
> ***
>
> We hope the explanations could address your concerns and we would appreciate it a lot if you could recognize the contributions of our work.
>
> [1] Huang G, Chen D, Li T, et al. Multi-scale dense networks for resource efficient image classification[J]. arXiv preprint arXiv:1703.09844, 2017.
> [2] Yang L, Han Y, Chen X, et al. Resolution adaptive networks for efficient inference[C]//Proceedings of the IEEE/CVF conference on computer vision and pattern recognition. 2020: 2369-2378.
> [3] Wang Y, Lv K, Huang R, et al. Glance and focus: a dynamic approach to reducing spatial redundancy in image classification[J]. Advances in Neural Information Processing Systems, 2020, 33: 2432-2444.
> [4] Wang Y, Huang R, Song S, et al. Not all images are worth 16x16 words: Dynamic transformers for efficient image recognition[J]. Advances in neural information processing systems, 2021, 34: 11960-11973.
> [5] Chen M, Lin M, Li K, et al. Cf-vit: A general coarse-to-fine method for vision transformer[C]//Proceedings of the AAAI Conference on Artificial Intelligence. 2023, 37(6): 7042-7052.
> [6] Han Y, Han D, Liu Z, et al. Dynamic perceiver for efficient visual recognition[C]//Proceedings of the IEEE/CVF International Conference on Computer Vision. 2023: 5992-6002.

---

> > ### Author Response · Authors · 2024-08-12
> > **Kind Reminder of Deadline**
> >
> > Dear Reviewer,
> >
> > Thanks for your efforts so far in our paper review.
> >
> > Currently, most of the concerns from other Reviewers have been well-addressed and we are eager to know whether our response has resolved your concerns. Concretely, we have added the discussion between Scala and dynamic networks which shows that our work is agnostic with this line of research. Moreover, we have clarified the contribution of our work and further expanded the application scenarios of Scala to foundation models like DINOv2.
> >
> > Due to the coming discussion deadline, we would like to kindly remind the Reviewer if our response through the whole rebuttal has addressed any of your concerns and helped the Reviewer to reevaluate this work.
> >
> > We really appreciate it if you could give us any feedback and your opinions are rather important to us. Thank you very much for your time!
> >
> > Sincerely,
> > Authors

---

> ### Comment · Area_Chair_sdRk · 2024-08-13
>
> Dear reviewer, please have a look at the rebuttal and indicate whether it addressed some of your concerns.

---

> ### Comment · Reviewer_7ef6 · 2024-08-14
>
> I highly appreciate the authors' response. However, I still think the lack of discussions and comparisons with dynamic networks in the original submission is an important limitation. Moreover, although the authors have provided some clarification, Isolated Activation seems like simply an engineering trick, and the proposed knowledge distillation method seems too incremental compared to existing works. These techniques may be insufficient as significant scientific contributions.
>
> I increase my rating to 4. However, I still vote to reject this paper. The major reasons are (1) lack of discussions on important related works, and (2) weak technical novelty.

---

### Official Review · Reviewer_V4fU · 2024-07-12

**Soundness:** 3
**Presentation:** 4
**Contribution:** 3
**Rating:** 7
**Confidence:** 3

**Summary:**

The paper presents Scala, a novel framework for scalable representation learning developed from US-Net. It identifies the issues of directly applying US-Net to ViTs and proposes solutions including Isolated Activation, Scale Coordination, and Stable Sampling. These innovations enable Scala to output several sub-networks in one-shot learning. Extensive experiments on various network architectures and datasets demonstrate that the sub-networks produced by Scala consistently outperform those generated by separate training, with significantly reduced training time.

**Strengths:**

Originality: Scala addresses the limitations of US-Net and successfully applies the concept of scaling to ViT backbones. This is a significant step in the adaptation of scaling methods for more complex network architectures.

Quality: The paper supports its claims with extensive experimental results, providing strong evidence for the effectiveness of Scala.

Clarity: The paper is clearly written and well-organized, making it accessible and easy to follow.

Significance: Scala has the potential to influence future research directions in scaling ViTs.

**Weaknesses:**

Originality: The novelty of Scala is somewhat constrained. For instance, Noise Calibration does not show a distinct difference from standard knowledge distillation. Essentially, Scala integrates US-Net with an alternative activation for the smallest subnet and fixed scaling ratios.

Quality: The authors might consider emphasizing results from a more standard 300-epoch ViT training schedule to align with common practices in the field.

Clarity: No further issues.

Significance: The challenge of scaling ViTs with arbitrary ratios remains unresolved.

**Questions:**

1. Regarding the issue shown in Fig. 4, is it always the smallest subnet that causes the issue, or does it occur with subnets having a scaling ratio near 0.25?

2. Can the authors clarify any differences between Noise Calibration and standard knowledge distillation?

3. What would be the impact if the distillation part were discarded and only Cross-Entropy (CE) loss were used for the initial epochs?

**Limitations:**

The novelty and significance of Scala are somewhat limited, as discussed in the weaknesses section. However, the extensive experimental results provide a robust foundation for the claims made in the paper. Overall, the work is well-executed and makes a valuable contribution to the field, justifying a recommendation for weak acceptance.

---

> ### Author Rebuttal · Authors · 2024-08-07
>
> We appreciate the Reviewer's approval and valuable comments. We respond to the Reviewer's concerns as below.
>
> ***
>
> **Weakness 1: Fixed scaling ratio.**
>
> Thank you for the comment. The hidden dimension of ViT has to be an integer multiple of the number of heads (e.g., 6/12) so ViT cannot support arbitrary width ratios like CNN. For instance, the theoretical maximum number of networks that ViT-B could represent is 25 if $s=0.25$. However, we do acknowledge that Scala still cannot generalize to unseen width ratios as this issue is essentially related to the architecture design of transformer and we constrain ourselves from modifying the conventional structure as it has been well-optimized in various scenarios. On the other hand, it further emphasizes that the problem we target to solve is very challenging as the intermediate subnets ($X=13$) are only trained for around 18 epochs and they are expected to match the performance of networks that are trained for 100 epochs.
>
> **Weakness 2: Results of 300 epochs.**
>
> Thanks for the advice and we will emphasize the 300-epoch results of Sec. 5.3 and 5.4 in the final version.
>
> **Question 1: Root cause.**
>
> Thanks for the great question. It is the smallest subnet that causes the issue no matter what the width is and we validate it by setting $s=0.5$ to be the smallest scaling ratio and removing Isolated Activation. The table shows that constantly activating the smallest subnet in the naive way still results in worse performance even if the width ratio is larger than 0.25.
>
> | Method  | $r=0.50$ | $r=0.75$ | $r=1.00$ |
> | :-----: | ------------ | ------------ | ------------ |
> | Scala ($s=0.5$) | 71.2% | 74.5% | 76.8% |
> | Scala ($s=0.5$) w/o IA | 70.2% | 73.1% | 75.9% |
>
>
> **Question 2: Noise Calibration.**
>
>
> Compared to standard knowledge distillation, the teacher of those subnets in Scala is dynamically changing (different widths) and will be optimized during training like the students as their weights are shared. On the one hand, the intermediate teachers serve as teacher assistants which fill the gap between $F^{l}\left (\cdot  \right )$ and $F^{s}\left (\cdot  \right )$ and provide simplified optimization objective for small subnets, in contrast to US-Net which utilizes $F^{l}\left (\cdot  \right )$ as the teacher for all subnets and results in inferior performance. On the other hand, it also indicates that those teachers cannot provide faithful predictions at the early stages and it is essential to add Cross-Entropy loss to those subnets during training which is overlooked in previous baseline methods.
>
>
> **Question 3: Impact of only CE at early epochs.**
>
> Thanks for the valuable suggestion. We further conduct experiments by removing the distillation loss of subnets in the first 10 epochs (denoted by *) and there is only minor performance change in the subnets which suggests that Noise Calibration is indeed helpful at early stages.
>
> | Method  | $r=0.25$ | $r=0.50$ | $r=0.75$ | $r=1.00$ |
> | :-----: | ------------ | ------------ | ------------ | ------------ |
> | Scala | 58.7% | 68.3% | 73.3% | 76.1% |
> | Scala* | 58.8% | 68.1% | 73.6% | 76.1% |
>
>
> ***
>
> We hope the explanations could address your concerns and we would appreciate it a lot if you could recognize the contributions of our work.

---

> > ### Comment · Reviewer_V4fU · 2024-08-13
> >
> > Thank you for the detailed response. I have updated my vote to accept.

---

> > > ### Author Response · Authors · 2024-08-13
> > > **Thanks for the support**
> > >
> > > Dear Reviewer,
> > >
> > > Thank you so much for the constructive comments for us to improve the work and we sincerely appreciate your support.
> > >
> > > Thanks again for being with us so far.
> > >
> > > Sincerely,
> > > Authors

---

### Official Review · Reviewer_bSkR · 2024-07-14

**Soundness:** 3
**Presentation:** 3
**Contribution:** 3
**Rating:** 5
**Confidence:** 3

**Summary:**

The paper introduces Scala, a novel framework designed to effectively scale down Vision Transformers (ViTs) for use in environments with fluctuating resource constraints. The key insight is that smaller ViTs can function as sub-networks within a larger ViT, differing mainly in width. Scala enables a singular network architecture that can emulate multiple smaller ViTs, thereby offering versatile inference capabilities while maintaining the structural principles of ViTs. The framework uniquely incorporates multiple sub-networks during its training phase, utilizes Isolated Activation to differentiate the smallest sub-network, and implements Scale Coordination to streamline the learning objectives for each sub-network, aiming for simplicity, stability, and accuracy. The empirical results across various tasks confirm that Scala can learn scalable representations efficiently with a single training iteration, maintaining the integrity of the original ViT architecture and achieving performance on par with networks trained separately.

**Strengths:**

The proposed Scala framework aims to enhance Vision Transformers (ViTs) by enabling them to learn scalable representations suitable for flexible inference. This is achieved through two key innovations: Isolated Activation, which effectively disentangles the representation of the smallest subnet to maintain clarity and specificity, and Scale Coordination, which ensures that each subnet within the larger network receives simplified, consistent, and accurate signals. These mechanisms are designed to optimize the performance and scalability of ViTs, addressing common challenges in adapting these architectures to varied and dynamic operational contexts.

**Weaknesses:**

1. Recent papers[1,2,3] with "Scalable" usually scale ViT to billion size with large scale datasets like DFN, JFT, and Datacomp. Therefore, I suggest authors should reconsider if the experiments can support "Scalable".


[1] Zhai, X., Kolesnikov, A., Houlsby, N., & Beyer, L. (2022). Scaling vision transformers. In Proceedings of the IEEE/CVF conference on computer vision and pattern recognition (pp. 12104-12113).

[2] El-Nouby, A., Klein, M., Zhai, S., Bautista, M. A., Toshev, A., Shankar, V., ... & Joulin, A. (2024). Scalable pre-training of large autoregressive image models. arXiv preprint arXiv:2401.08541.

[3] Dehghani, M., Djolonga, J., Mustafa, B., Padlewski, P., Heek, J., Gilmer, J., ... & Houlsby, N. (2023, July). Scaling vision transformers to 22 billion parameters. In International Conference on Machine Learning (pp. 7480-7512). PMLR.

**Questions:**

please refer the weakness.

**Limitations:**

This paper addresses the limitations in conclusion

---

> ### Author Rebuttal · Authors · 2024-08-07
>
> We appreciate the Reviewer's comments to point out the confusing description and we make the response as below.
>
> ***
>
> **Weakness 1: Phrase.**
>
> Thank you for your insightful suggestion. We will revise our presentation in the final version to prevent any potential misunderstanding. Our motivation for adopting the term "scalable representation learning" stems from the observation that different methods (e.g., DeiT, MAE, DINOv2) learn distinct representations that are not inherently slimmable (see Appendix: A.3). Scala is proposed to learn the original representations while incorporating slimmability. Therefore, we initially chose "scalable representation learning" to describe our method. However, we acknowledge that the term "scalable" may lead to confusion, and we will revise it in the final version to enhance clarity.
>
> ***
>
> We hope the explanations could address your concerns and we would appreciate it a lot if you could recognize the contributions of our work.

---

### Official Review · Reviewer_rLHb · 2024-07-17

**Soundness:** 3
**Presentation:** 2
**Contribution:** 2
**Rating:** 6
**Confidence:** 3

**Summary:**

This paper advances an approach for training Vision Transfomers (ViTs) such that at inference time they can be dynamically adjusted to fit different budget constraints with reduced drops of performance. To this end, the authors introduce Scala, a framework that allows a single network to encapsulate and train simultaneously multiple sub-networks of different capacities and widths. The methodological backbone of this work are the Universally slimmable networks (US-Net) [37], originally devised for CNNs. The authors identify and analyze a few flaws of US-Nets: difficulty to generalize to ViTs, small interpolation and extrapolation ability to sub-network size unseen during training, impact of sustained activation of the smallest sub-network that coupled with the sandwich rule for selecting sub-networks during training leads to an over-emphasis on it at the expense of the other sub-networks.
The authors propose two simple strategies towards such a method for ViTs: (i) Isolated activation that separates the smallest sub-network from the other sub-networks; (ii) scale coordination consisting of a set of heuristics to ensure that each sub-network gets simple, accurate and stable learning objectives: (a) progressive knowledge transfer from larger networks to smaller ones in gradual decrease of capacity, (b) stable sampling of intermediate width ratios to avoid large variations in capacities in the sandwich, (c) noise calibration, essentially a composite loss of supervised cross-entropy and distillation from the bigger sub-network.
Scala is evaluated on several settings on the ImageNet-1k dataset with ViT-Ti/S/B, hybrid CNN-ViT architectures, lightweight networks, but also for dense prediction on semantic segmentation and self-supervised pre-training with interesting results. The baselines used here were Separate Training,  Autoformer and US-Net.

**Strengths:**

### Significance
- the paper deals with a challenging and useful task for deploying ViT models into different operational settings with different computational constraints without retraining or distilling specific architectures each time

- although a computational overhead is expected for such methods, the main components of Scala are relatively simple and make sense

- Scala achieves good performance with a higher boost in the low parameter regime

### Originality
- the proposed contributions are somehow incremental as they are improving the US-Net prior work, but do have some novelty and they are simple.

### Clarity
- in general this work is well argued and easy to follow. The authors construct well the arguments regarding the challenges when going from CNNs to ViT with US-Net and how to construct their Scala approach.

### Quality
- the paper offers several experiments and studies in the main paper and in the appendix (longer training, fast interpolation, ablation of components) that are well thought and improve the understanding of the method.

- I appreciate the experiments beyond image classification, on semantic segmentation, as well as the self-supervised pretraining and subsequent linear probing on a downstream task.

**Weaknesses:**

### "Scalable" naming
- I think that the framing of the method as _"scalable representation learning"_ is quite confusing as it's not representative for this task, it's not a name used by other related works. Importantly, it can be easily mistaken with most works that use "scalable" for depicting the ability/property of a system (method, architecture) to handle a growing amount of data, parameters, and the potential to accommodate this growth. In other words "scalable" is rather used for depicting scaling up, whereas this work depicts the property of the proposed approach to accommodate sub-networks of different lower sizes/scales from the original.

- maybe other names us in related works would be more appropriate here: slimmable, elastic, modular, flexibile inference, etc.


### Limited baselines and related work
- some relevant related works dealing with tranformer networks are either just briefly mentioned, e.g., Matformer  [18], or not mentioned at all, e.g., SortedNet [a], Early exit [b]

- One of the main baselines, US-Net is originally designed for CNNs and, as the authors mentioned, moving to ViTs is not straightforward. Matformer is criticized for the limited number of models produced, but can be considered in the several experiments with X=4 sub-networks. Matformer and SortedNet could be included in the experimental evaluation


### Scope of experiments
- While the authors considered several settings for computer vision tasks (image classification, segmentation, light architectures), transformer architectures are also encountered in NLP (as mentioned by the authors in L56). In such cases the original models can have much more parameters and elastic inference for lower computational budgets would be of high interest.

- It would be useful to include an experiment from NLP in the style of those from Matformer or SortedNet.

- The biggest architectures used here is a ViT-B (~86M params). Extending experiments to larger modern architectures would be definitely useful and interesting.

### Clarity
- it's not always clear in the text and cost estimations that Scala needs a pre-trained full network as teacher for the distillation. This add some cost in compute and time in the end. Besides it's not clear whether US-Net also needs and uses a pre-trained teacher in the reported results.

- in the intro, the authors mention that they address the issue of minimal interpolation ability of ViTs. Results from Table 2 show that the interpolation abilities of ViTs with Scala are still very low. However the fast interpolation strategy from $\S$A.2 is actually interesting for practical settings even though not fully solving this issue. It might be worth moving up in the main paper.

- the idea of the transferability experiment ($\S$5.4) with DINOv2 is nice. From the description it is not clear whether DINOv2 was used as teacher for the distillation or also as supervised pre-training on ImageNet-1k? Or the pre-training on ImageNet-1K was done in a supervised manner as in previous experiments?

- the ablation experiment from Table 6 is nice. However the presentation with removing one component at once offers only a partial understanding of the contributions of each module. Different configurations with different modules in on/off mode should give a better global understanding.



**References:**

[a] Valipour et al., SortedNet: A Scalable and Generalized Framework for Training Modular Deep Neural Networks, arXiv 2023

[b] Xin et al., Deebert: Dynamic early exiting for accelerating bert inference, ACL 2020

**Questions:**

This paper takes an interesting direction of study: how to train ViTs such that a fine-grained elasticity in terms of sub-network sizes is possible at runtime.
The proposed Scala approach is well described, makes sense and achieves good results in several computer vision settings.

I do have a few concerns related to the phrasing of this type of works ("scalable representation learning") which can be confusing, the absence of larger architectures and of recent relevant baselines.
My current rating is mildly positive (rather on the fence though) and I'm looking forward for the rebuttal.

Here are a few questions and suggestions that could be potentially addressed in the rebuttal or in future versions of this work (please note that suggested experiments are not necessarily expected to be conducted for the short rebuttal period):

1. Please clarify the points raised in the clarity section: use of teacher model for Scala and US-Net, implementation of transferability experiment.

2. Comparison of training cost between Scala (including teacher training), US-Net and Separate Training baselines.

3. Add a discussion of differences and when possible experimental comparison with Matformer and SortedNet baselines on image classification or semantic segmentation.

4. Extension of experiments to NLP architectures and tasks in the style of SortedNed, Matformer

**Limitations:**

The authors addressed some of the limitations in the conclusion section.

---

> ### Author Rebuttal · Authors · 2024-08-07
>
> We sincerely appreciate the Reviewer’s detailed comments and constructive suggestions for us to improve our work. We make the response as below.
>
> ***
>
> **Weakness 1: Phrase.**
>
> Thanks for the great suggestion and we will modify the presentation in our final version. Our motivation for adopting 'scalable representation learning' is that different methods (DeiT, MAE, DINOv2) will learn different representations, which are not naturally slimmable (see Sec A.3). Scala is proposed to learn the original representations but with the slimmable ability, so we adopt 'scalable representation learning' to describe our method. However, we do agree that the term 'scalable' may result in misunderstanding and we will modify it in the final version.
>
> **Weakness 2: Limited baselines.**
>
> Thanks for the comment.
>
> SortedNet scales the network through multiple directions and results in irregularities in transformer architectures, compared to Scala which aligns with the inherent design of ViT to vary from widths. As their vision experiments are conducted on small-scale CIFAR datasets, we intended to reproduce their method on ImageNet but did not find released implementations.
>
> MatFormer only scales the FFN block in transformer so it offers a minor computational adjustment space and we adapt their method (written in JAX) on DeiT-S for a fair comparison. The results are shown in Fig.1 of the PDF file (reply to all reviewers) and Scala achieves comparable performance with it (better in most cases) when $s=0.5$ with a larger adjustment scope.
>
>
> **Weakness 3: Scope of experiments.**
>
> We thank the Reviewer for the valuable suggestion. Indeed, extending Scala to NLP and a larger network is part of our original plan but we are struggling with computation resources as both of the experiments demand much larger training costs. We will definitely consider it as our future plan.
>
> **Weakness 4: Teacher and training costs.**
>
> As explained in Weakness 1, Scala can be regarded as a tool to make existing methods become slimmable while trying to maintain the original representations.
> Although we can expect to learn similar representations of methods trained with supervised learning on ImageNet-1K (DeiT), some self-supervised learning methods (DINOv2) are trained with gigantic undisclosed data which is almost impossible to reproduce their performance without utilizing the pre-trained model.
> So we simply use the pre-trained network (same size as the student) as the teacher to imitate the original model to inherit the original representations, instead of utilizing a larger network which usually leads to better performance.
>
> For US-Net, we do not introduce teacher networks following their original design but we do compare with Separate Training with Distillation in Sec A.8 (Tab. 10), where Scala still outperforms the baseline at smaller widths. Here we further re-implement US-Net by adding the teacher and the results are shown in the table below. Although obtaining a better full model, US-Net exhibits worse performance at smaller width ratios because it utilizes the full network as the teacher for all subnets and this phenomenon verifies our motivation of proposing Progressive Knowledge Transfer.
>
>
> | Method  | $r=0.25$ | $r=0.50$ | $r=0.75$ | $r=1.00$ |
> | :-----: | ------------ | ------------ | ------------ | ------------ |
> | US-Net | 52.7% | 60.1% | 66.9% | 73.2% |
> | US-Net+KD | 51.6% | 58.7% | 66.6% | 73.8% |
> | ST | 45.8% | 65.1% | 70.7% | 75.0% |
> | Scala | 58.7% | 68.3% | 73.3% | 76.1% |
>
> Assuming to deliver 13 models in the end, we compare the training time (100 Epoch) of Scala with US-Net, Separate Training, and Separate Training with Distillation on 8 A100 GPUs and we do not include the teacher training time as Scala is aimed to scale down existing works where the same size pre-trained model can be easily downloaded. The difference between US-Net and Scala is not large as the transformer architecture has been well-optimized on GPU and we do observe a significant time gap between Scala and ST/STD as they have to train 13 models iteratively. Moreover, Scala can be configured to deliver 25 models without an increase in training time as we sample 4 networks at each iteration in all scenarios which further highlights our strengths.
>
> | Method  | Training Hours |
> | :-----: | ------------ |
> | ST | 123 |
> | STD | 128 |
> | Scala | 21 |
> | US-Net | 20 |
>
> **Weakness 5: Interpolation.**
>
> We did not fully address the interpolation problem as this issue is essentially related to the architecture design of transformer and we constrain ourselves from modifying the conventional structure as it has been well-optimized in various scenarios. On the other hand, it further emphasizes that the problem we target to solve is very challenging as the intermediate subnets ($X=13$) are only trained for around 18 epochs and they are expected to match the performance of networks that are trained for 100 epochs.
>
> Thanks for carefully reading our paper and we will move Fast Interpolation into the main paper.
>
> **Weakness 6: DINOv2.**
>
> We follow the previous setting except for using DINOv2-B as the teacher which means we still train the model in a supervised manner. We have tried to adopt the self-supervised learning objective from DINOv2 but found the training cost too large, and it is almost impossible to reproduce the performance following the original learning objective with ImageNet-1K since DINOv2 is trained on the private dataset LVD-142M. However, we re-conduct the experiment by removing all the CE loss to alleviate dataset bias and find this choice results in surprising generalization ability by inheriting the fruitful knowledge of DINOv2 (please see the reply to all reviewers).
>
> **Weakness 7: Ablation.**
>
> Thanks for the advice and we will add more ablation in our final version.
>
> ***
>
> We hope the explanations could address your concerns and we would appreciate it a lot if you could recognize the contributions of our work.

---

> > ### Comment · Reviewer_rLHb · 2024-08-12
> > **Post-rebuttal comments**
> >
> > I would like to thank the authors for the detailed and informative rebuttal. I imagine they invested significant effort into clarifying the concerns from the 4 reviewers and I'm confident they will improve this work.
> >
> > I've read the other reviews and the responses of the authors to them and skimmed through the paper again.
> >
> > The strong points I see in this rebuttal are:
> > - several clarifications around the implementation, in particular on the use of pre-trained teacher across settings: both on the regular setting but also on the transferability task where DINOv2 was used as teacher and not as a self-supervised objective
> > - the addition of a comparison with a more recent and related work, Matformer. Pity that the code of SortedNet is not available.
> > - more detailed information about training costs w.r.t. to the main baseline US-Net, that originally does not benefit from a pretrained teacher. The authors report results for US-Net + KD in the rebuttal and it still under-performs compared to Scala.
> > - new results using DINOv2 as teacher without cross-entropy loss showing the effectiveness of the option of using foundation models for this setting.
> >
> > On the downside the evaluation is still limited to mid- to small-sized architectures (the biggest architecture used here is a ViT-B  with 86M parameters) and to computer vision settings. I can relate to the argument of the authors regarding computational cost. However given their results with DINOv2, one may devise a similar strategy for distilling a NLP foundation model to a different task than next-token prediction, e.g., text-classification. Hopefully the authors will find a solution in this direction for the paper update.
> >
> > **Wrap-up**. I think the submission has improved over the rebuttal with the new information and results and I encourage the authors to include the new findings in the main paper, and to release implementation code (as they stated in the NeurIPS paper checklist, to help other works to build upon and compare against Scala). At this point I'm rather leaning for a positive recommendation for this work. I do not have other questions.

---

> > > ### Author Response · Authors · 2024-08-12
> > > **Thanks for the support**
> > >
> > > Dear Reviewer,
> > >
> > > Thank you very much for your recent feedback. We truly appreciate your support and the time you have invested in reviewing our paper. Yes, we will definitely include the new findings in the main paper and we promise to release the code if this work is accepted.
> > >
> > > We are glad to hear that the reviewer has no concerns and is leaning toward a positive recommendation. If the reviewer finds that the work is improved during the rebuttal, we would be grateful if you could reflect the updated stance in the rating on OpenReview. This would greatly assist us in the final decision-making process.
> > >
> > > Thank you once again for your thoughtful consideration and valuable insights.
> > >
> > > Sincerely,
> > > Authors

---

### Author Rebuttal · Authors · 2024-08-07

Dear Reviewers:

Thanks for your valuable comments in the review process. We have an exciting experiment added during rebuttal which supports that Scala can effectively inherit the generalization ability from foundation models like DINOv2 while maintaining the flexible inference capability. This indicates that it may be possible to enable foundation models to become slimmable as well which would further enhance the application scenarios of our method.

Inspired by the comment of Reviewer rLHb, we build Scala over DeiT-B for ImageNet-1K training and utilize DINOv2-B as the teacher in order to inherit its strong generalization ability. However, we remove all the Cross-Entropy loss during training to alleviate the dataset bias issue as DINOv2 is trained on the gigantic private dataset. To examine the generalization ability of Scala, we conduct linear probing on 12 fine-grained classification datasets following DINOv2. Tab. 1 in the PDF file shows that Scala significantly outperforms DeiT variants on the average performance of fine-grained classification which suggests that Scala indeed inherits the fruitful knowledge from DINOv2 with remarkable improvement in its generalization ability. Moreover, the improvement over DeiT does not decrease when we scale down the width ratios during inference and it indicates that Scala maintains the flexible inference capability very well even though it contains more knowledge than before.

We thank the Reviewers for the constructive suggestions which help us to improve the work. We are actively available until the end of this rebuttal period and please let us know if you have any further questions. Thank you so much for being with us so far.

Sincerely,
Authors

---

### Author Response · Authors · 2024-08-11
**Deadline Approaching**

Dear Reviewers:

Thank you so much for delivering your valuable comments to us.

We have tried our best to provide a detailed response based on your concerns. We would like to kindly remind you that Aug 13th is the deadline for the discussion phase and your opinions regarding our response matter a lot to us.

Thank you so much for helping us improve our paper.

Sincerely,
Authors

---

> ### Comment · Area_Chair_sdRk · 2024-08-13
>
> Dear authors, thank you for your rebuttal. FYI: after the author discussion phase closed, there is still a bit over a week of discussion between ACs and Reviewers.

---

### Decision · Program_Chairs · 2024-09-25

**Decision:**

Accept (poster)

**Comment:**

After rebuttal and discussions, reviewers generally lean towards acceptance. I do agree with the originally more conservative reviewers that the related work is not discussed enough. For two examples: while MatFormer is discussed, there are several more works from Aditya Kusupati that are highly relevant. Furthermore, FlexiViT tackles all the same issues but along sequence length.

Thus, for the general case of "flexible amount of computation at inference time", too few baselines are compared to. That being said, the presented observations and introduced ideas are interesting and novel enough that, compared to their obvious baselines/ablations, they are worth publishing. Hence, we still recommend acceptance.

Finally, the AC agrees with reviewer rLHb that the title is both much too broad, and a wrong use of the word "scalable" compared to its typical meaning within the community. Not only that, also "representation learning" is not a good fit for this as it usually has a different meaning/focus. Please change the title to something more fitting and commong for this scenario. Some examples could be "Elastic training of Vision Transformers" or "Vision Transformers with Flexible Width" or something to that effect.